# "I'm Afraid I Can't Do That, Dave"; Getting to Know Your Buddies in a Human–Agent Team

**Maarten P. D. Schadd** [1,*], **Tjeerd A. J. Schoonderwoerd** [2], **Karel van den Bosch** [2], **Olaf H. Visker** [1], **Tjalling Haije** [2] and **Kim H. J. Veltman** [2]

[1] TNO, Oude Waalsdorperweg 63, 2597 AK The Hague, P.O. Box 96864, 2509 JG Den Haag, The Netherlands; olaf.visker@tno.nl
[2] TNO, Kampweg 55, 3769 DE Soesterberg, P.O. Box 23, 3769 ZG Soesterberg, The Netherlands; tjeerd.schoonderwoerd@tno.nl (T.A.J.S.); karel.vandenbosch@tno.nl (K.v.d.B.); tjalling.haije@tno.nl (T.H.); kim.veltman@tno.nl (K.H.J.V.)
* Correspondence: maarten.schadd@tno.nl

**Abstract:** The rapid progress in artificial intelligence enables technology to more and more become a partner of humans in a team, rather than being a tool. Even more than in human teams, partners of human–agent teams have different strengths and weaknesses, and they must acknowledge and utilize their respective capabilities. Coordinated team collaboration can be accomplished by smartly designing the interactions within human–agent teams. Such designs are called Team Design Patterns (TDPs). We investigated the effects of a specific TDP on proactive task reassignment. This TDP supports team members to dynamically allocate tasks by utilizing their knowledge about the task demands and about the capabilities of team members. In a pilot study, agent–agent teams were used to study the effectiveness of proactive task reassignment. Results showed that this TDP improves a team's performance, provided that partners have accurate knowledge representations of each member's skill level. The main study of this paper addresses the effects of task reassignments in a human–agent team. It was hypothesized that when agents provide explanations when issuing and responding to task reassignment requests, this will enhance the quality of the human's mental model. Results confirmed that participants developed more accurate mental models when agent-partners provide explanations. This did not result in a higher performance of the human–agent team, however. The study contributes to our understanding of designing effective collaboration in human–agent teams.

**Keywords:** artificial intelligence; human–agent teams; human–AI teams; human–machine teaming; mental models; team design pattern; team collaboration

## 1. Introduction

The increasing development of Artificial Intelligence (AI) and technological innovations are changing the way individuals and teams learn and perform their tasks. In hybrid teams, people collaborate with artificially intelligent partners (from now on: agents) to achieve a common team goal. An important question regarding teamwork in hybrid teams is how its members can be adequately supported to establish coordinated task operation. It is believed that effective collaboration can be accomplished by smartly designing the interaction between hybrid team partners [1,2].

In expert human teams, members use their knowledge about the relationship between task demands, competencies of team members, and situational circumstances. By sharing this awareness, expert teams are able to assign tasks to the members in a flexible and dynamic manner, allowing the team to achieve a better overall performance [3,4]. Achieving such aligned collaboration in human-only teams is by no means self-evident, and requires thoughtful design and deliberate practice [5,6]. One reason why human-only teams have

the potential to evolve into an expert team is that members share the same systems for information processing. Each member is aware that the others in the team perceive, reason, and think in more or less the same terms as they do. In hybrid teams though, humans and agents have different information processing systems that underlie their intelligence. Hence, achieving common ground and aligning task objectives sets additional demands [7]. One distinctive difference is that in a hybrid team, the team members are a priori not familiar with the capabilities and restrictions of each other's systems. Regardless, in order to understand each other, which is a prerequisite for being able to coordinate and cooperate, the interactions between humans and agents have to be explicitly designed to bring about experiences that enable team members to improve their understanding of the task and of their teammates. Furthermore, building upon the acquired understanding, interaction designs for hybrid teams should guide partners how to select the strategies that support the overall performance of the team. Thus, the purpose of an interaction design is to guide a team with proven solutions on how to operate when it is faced with a specific and repeatedly occurring task situation or problem [8].

Recently, it has been proposed to use team design patterns (TDPs) as an approach for designing human–machine teaming [8–10]. There have been studies into the design of such patterns [11], and also in how TDPs affect team functioning and team performance [12,13]. A common feature of TDPs is that they require team members to have an internal or mental representation about the task and the team. Otherwise members would lack the necessary knowledge for executing the collaborative interactions as intended [14]. However, TDPs do not only require a basic internal representation, their execution should also bring forth experiences that enable partners to improve and refine their mental representations, thus enhancing the quality of teamwork in the long term. An important question is whether the experiences evoked by a TDP are by themselves sufficient for team members to develop and expand their mental models, or that additional explanations are needed to achieve learning benefits [13,15,16].

In this paper we present a preliminary study into the effects of a team design pattern on the collaboration of agents in an agent-only team. In an experiment with human participants, we then investigate the effects of this TDP with and without agent-explanations on the collaboration and learning in a hybrid team. We used one specific pattern, proactive task reassignment, that guides members of a team in deciding whether or not to exchange assigned tasks. We first investigated whether agent teams that operate according to this pattern function and perform better than control teams. Then, we investigate if agents that provide explanations for their decisions yield benefits to the human team member in terms of knowledge about the task and the team, and on the performance of the team as a whole. This study amends our knowledge of how to design human–agent team collaboration in such a manner that partners get to know one another, and develop a shared understanding. Such common ground is of critical importance for a team [17]. The movie "2001: A Space Odyssey" shows what happens if there is no common ground. Upon commands of the scientist dr. David Bowman, the computer agent HAL 9000 just responds with: "*I'm sorry Dave. I'm afraid I can't do that*", leaving Bowman bewildered and frustrated. Thus, getting to know your teammates and developing and maintaining a shared objective and strategy is key in effective human–agent teaming.

## 2. Related Work

Teamwork is the process through which team members collaborate their task work to achieve team goals [18,19]. The development of intelligent agents to perform in a team is often focused on the task work, not the teamwork. As a result, tasks tend to be allocated to team members in a fixed manner. However, hybrid human–agent teams can be more effective when, just like in human expert teams, tasks can be assigned dynamically between all team members. This research proposes an approach for designing agents that use teamwork skills to support dynamic task allocation.

Patterns of interaction that prove to be successful for the team, and that are reusable in recurring similar situations, are called Team Design Patterns (TDPs) [10]. A team may use a variety of TDPs to successfully address different kinds of recurring problems. One such problem may concern how to assign the various task among the team members in such a way that it supports effective and efficient performance by the team as a whole. The TDP *proactive task reassignment*, here abbreviated as PATRA, was designed to guide a team addressing this problem. The objective of PATRA is to distribute the required work effectively by utilizing the different strengths and weaknesses of humans and agents. The pattern describes various actions of—and interactions between—team members that help to achieve this objective, such as dictating a low-skilled team member to request a higher-skilled team member to take over a task. For example, the TDP dictates team members with low workload to request tasks from busy team members. Such interactions might facilitate that the workload is divided more equally, thus supporting the team's overall effectivity and efficiency.

It should be noted that PATRA is not the only possible design solution for the task allocation problem. Some team tasks do not provide the right conditions for PATRA. If, for example, team members cannot access information about their team members, then PATRA is not suitable. In such a situation, a design in which the distribution of tasks among team members is allocated to one central leader may be more appropriate. As the suitability of a TDP depends on the nature of the task, the team, and the situational context, any TDP should, therefore, carefully specify under which conditions it should be applied (see also Table 1).

Team Design Patterns are proven solutions that help a team to organize their work and efforts in such a fashion that the team responds effectively to the demands of a particular type of task situation [10]. TDPs have in common that they require team members to have or obtain knowledge about all tasks and about all team members. The interactions described in TDP PATRA, in which team members delegate tasks among one another, require knowledge of:

- Competency (the skills of team members for each task);
- Workload (all team members' present and future tasks);
- Efficiency (the expected benefits in ratio to the required effort of the interaction).

If team members have this knowledge, then they have the information to establish whether it is beneficial for the team to either delegate a task, or to take over a task from another team member. Thus, TDPs might require team members to have mental models that include knowledge about the team's mission and the team's tasks, about their own role and tasks in the team, and of the role and tasks of the other team members.

There is evidence that having accurate mental models of each other's competences and shortcomings is vital for team performance [20,21] and for efficient and cooperative human–robot interaction [22,23]. When human individuals are collected to become a team, they generally have some basic knowledge about the context, the task, and the team. They develop, correct, and refine their mental models through experience and team communication [24,25]. How this process culminates in a mental model is difficult to determine precisely. In fact, humans themselves are often not consciously aware of what and how they learn from experiences [26]. Humans may unconsciously select features from a team task experience and use these to refine their mental model [27]. This implies that the human may also select characteristics that are not or only partially related to the task. For example, a human may notice that two team agents often follow each other, and may record in its mental model the deduction that the two are friends. This assumption may affect the human's behavior, perhaps without consciously realizing it. Or, another example, the human may notice that a particular team member has a preference for a certain location in the scenario. The examples point out that the specific information humans extract from experiences cannot be predicted exactly, or how they will use this to update their mental model. This is different when we consider the development of the knowledge representation of an agent [28]. When an agent is programmed to act as

member of a team, it too is initially programmed with basic knowledge about the context, the task, and the team. In contrast to humans, the way how agents develop, correct, and refine their knowledge representation from experiences is precisely defined in the computational source.

One objective of Team Design Patterns is to bring about experiences that potentially support human and agent team members to improve their internal models. Potentially, because learning from experiences is not guaranteed, as the meaning of an agent's act may be obscured to the partner agent. For example, suppose that one member requests another team member to take over its task. What exactly does it mean if the addressed team member rejects the request? Does it mean that the addressed agent feels it lacks the required competency to do the task? Or does the addressed agent think that yet another team member may be even more competent? Or does the addressed agent conclude that an additional task would aggravate its task-load too much? It is difficult for the requesting team member to tell what the reason is for the rejection, preventing the agent in improving its internal model. Thus, sometimes the experience alone provides insufficient information to learn. It has been shown that providing explanations are important for collaboration and to improve learning (e.g., [16,29,30]).

Explanations are important for human–agent teams, as agents tend to have no or very little background knowledge. They often fail to understand the meaning of a particular observation, which may be obvious to a human observer, thus missing the opportunity for the agent to learn [31]. Vice versa, agents that provide explanations for their actions are essential for humans to learn from experiences [32]. Generating an explanation requires access to some sort of internal representation about the task, the team, and the situational context. Humans have a mental model [33]; agents have an internal representation in the form of a computer model. The quality and usefulness of generated explanations are bounded by the accuracy and elaborateness of the internal model. In order for an agent to provide usable and effective explanations, it must continually determine the human's perspective, requiring an updated representation of its teammate, often referred to as a Theory of Mind [34].

## 3. Use-Cases

This section introduces the selected use-cases for the two experiments with the TDP PATRA. In both use-cases, we investigate the applicability of the TDP. Team members have different skill levels for the various tasks in the use-case, meaning that they are more or less capable of completing a task. In the blanket search use-case, there are only soft dependencies between team members, meaning that team members are generalists that are able to perform every task in the scenario. In the USAR use-case, we also introduce hard dependencies in the team, meaning that team members are specialists who can perform particular tasks very well, but rely on others to complete other types of tasks in the scenario. By applying the TDP PATRA in these two different use-cases, we obtain insight into the generalizability of this team design pattern for different team collaboration structures.

### 3.1. Blanket Search

In a naval military context, "blanket search" is a damage assessment and repair procedure for incidents and malfunctions on a ship. A team of naval personnel performs a round, and each member is assigned a route along a series of compartments. Incidents, such as a fire, need immediate counteraction. Malfunctions, such as a failing energy system are often solved after incidents have been resolved. Typically, specialized engineers are assigned to routes that contain systems within their area of expertise to ensure that malfunctions are solved effectively and efficiently. Figure 1 shows the simplified simulation of the blanket search task that we developed, involving team members that are able to solve every incident and malfunction, albeit with varying degrees of efficiency resulting from differing skill levels.

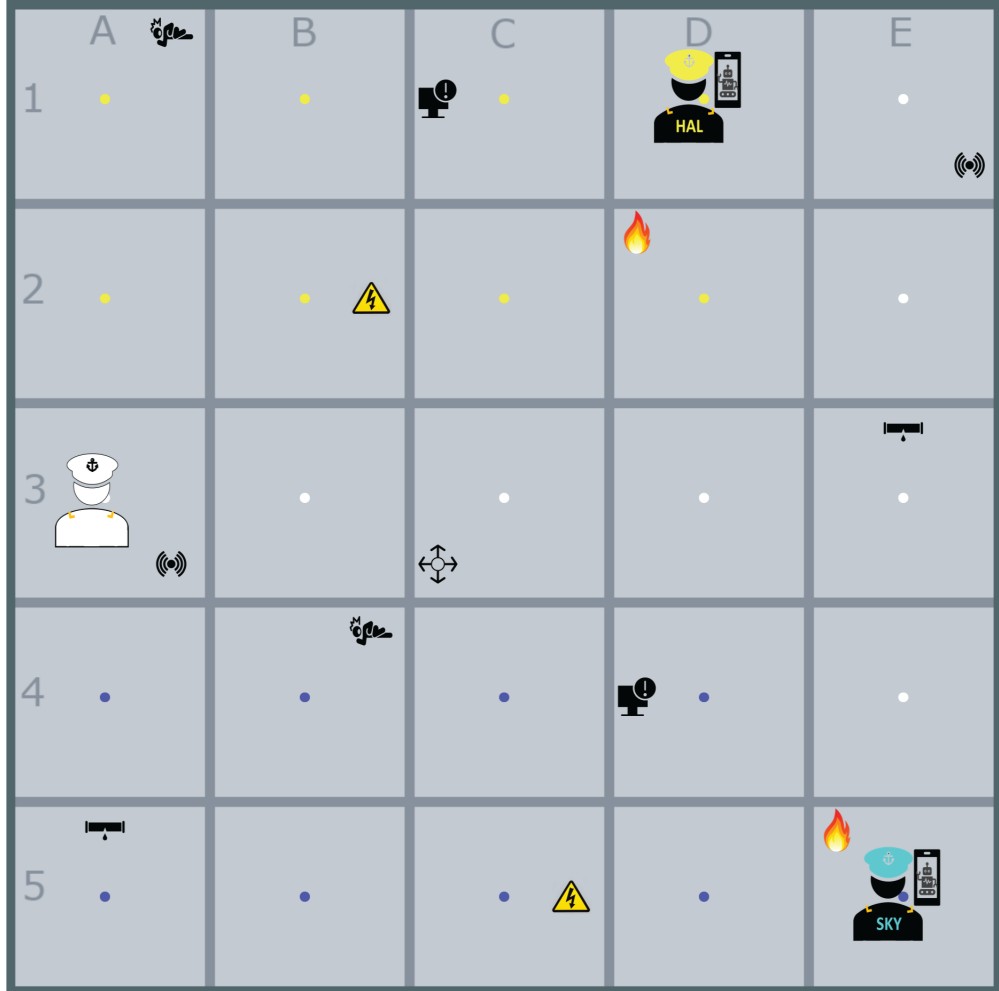

**Figure 1.** Screenshot of the simulated blanket search task, in which three team members collaborate to solve incidents (shown with icons representing fire, a leaking pipe, or wounded personnel) and malfunctions (shown with icons representing energy-, mobility-, SeWaCo-, and C4I-systems). Each cell represents a compartment, which belongs to a route of a particular agent. Routes are indicated by colored circles.

### 3.1.1. Objective and Tasks

The team's objective is to collectively complete a blanket search effectively and efficiently. This objective is achieved when all tasks (incidents and malfunctions) are found and repaired. Incidents and malfunctions both distinguish three levels of severity: mild, normal, and severe. The time that is required to solve a task increases with severity of that task. Repairing incidents has priority over repairing malfunctions. Three types of incidents were implemented in a simulated blanket search use-case: leakage, fire, and wounded personnel. In addition, malfunctions of four types of systems occurred during the scenarios: SeWaCo (sensor, weapon, and command), C4I (command, control, computer, communication, and intelligence), energy, and mobility systems.

### 3.1.2. Team Members

In both experiments, the blanket search team members all have different levels for the following skills:

- Extinguishing fires;
- Repairing leakages;
- Providing first aid to wounded personnel;
- Repairing SeWaCo malfunctions;

- Repairing C4I malfunctions;
- Repairing energy malfunctions;
- Repairing mobility malfunctions.

The difference in skill levels between members makes it useful for the team to learn about each other's capabilities, in order to effectively coordinate their task execution. The team conducts the blanket search according to the following protocol:

1. Each member has been assigned a specific route through the ship. As a team, they check all compartments;
2. Upon entering a compartment, any incidents appear automatically. Human and agent team members need to actively search (i.e., by executing a search action in the simulation) for any system malfunctions within each compartment. After performing a search, any found malfunctions appear;
3. Incidents on a route can only be solved by the team member who is assigned that route. When a team member finds a system malfunction on its route, it can also request another team member to perform the required repair.

Multiple incidents and/or malfunctions may be present in one compartment. If so, team members are programmed (in the case of agents) and instructed (in the case of human team members) to prioritize as follows:

- Extinguish fires;
- Repair leakages;
- Provide first aid to victims;
- Continue search;
- Handle malfunctions.

### 3.1.3. Agent Implementation

The agents are programmed to have a task model, including how repair time is affected by skill level. In addition, agents also have a mental model that includes information about the skill levels of the other agents. Furthermore, their mental models include the location of a task that was reassigned to them. The implemented behavior protocol for the autonomous agents is shown in Figure 2.

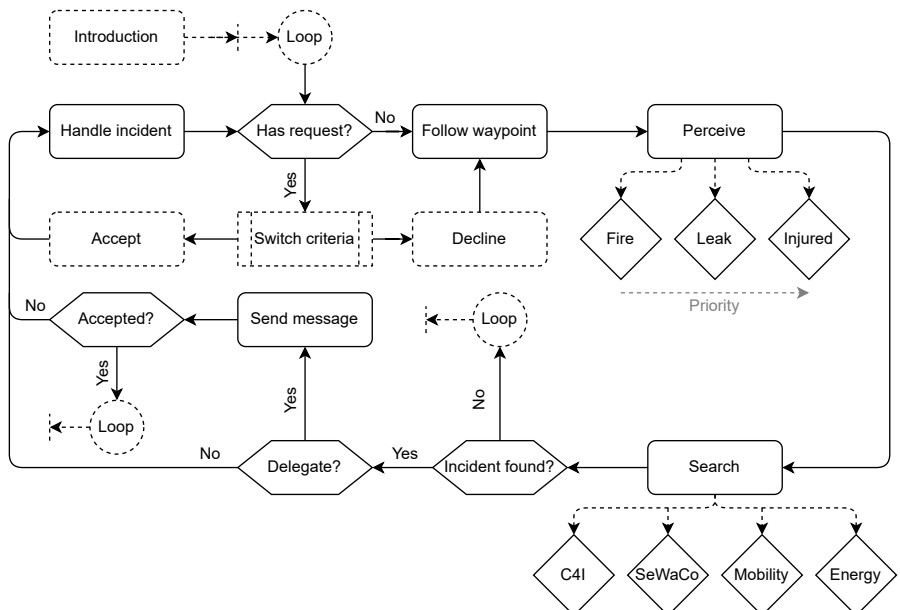

**Figure 2.** Behavior rules for autonomous agents in the blanket search scenario.

Before an agent proceeds to the next compartment of its route, it first checks whether it has received a request to take over a task of another team member. If so, it starts the procedure to consider whether to accept or to decline the request (see below). If not, it proceeds with its own assigned task.

Figure 3 shows the protocol for evaluating a task take-over request issued by another team member:

1. If no more scheduled tasks, then accept the request;
2. If for the requested task, the requesting agent's skill level exceeds the own skill level, then decline the request;
3. Else, if the distance from current location to the location of the requested task exceeds the threshold level, then decline the request;
4. Else, if number of scheduled tasks exceeds the number of scheduled task of the requesting agent by a certain threshold, then decline the request.

Otherwise, accept the task request. If the agent accepts a request, the agent moves to the task location and performs the repair. Whenever an agent finds a malfunction on its route, it will issue a task request if the difference between its own skill level and the estimated skill level of another team member (based on its knowledge about that team member) exceeds a particular threshold.

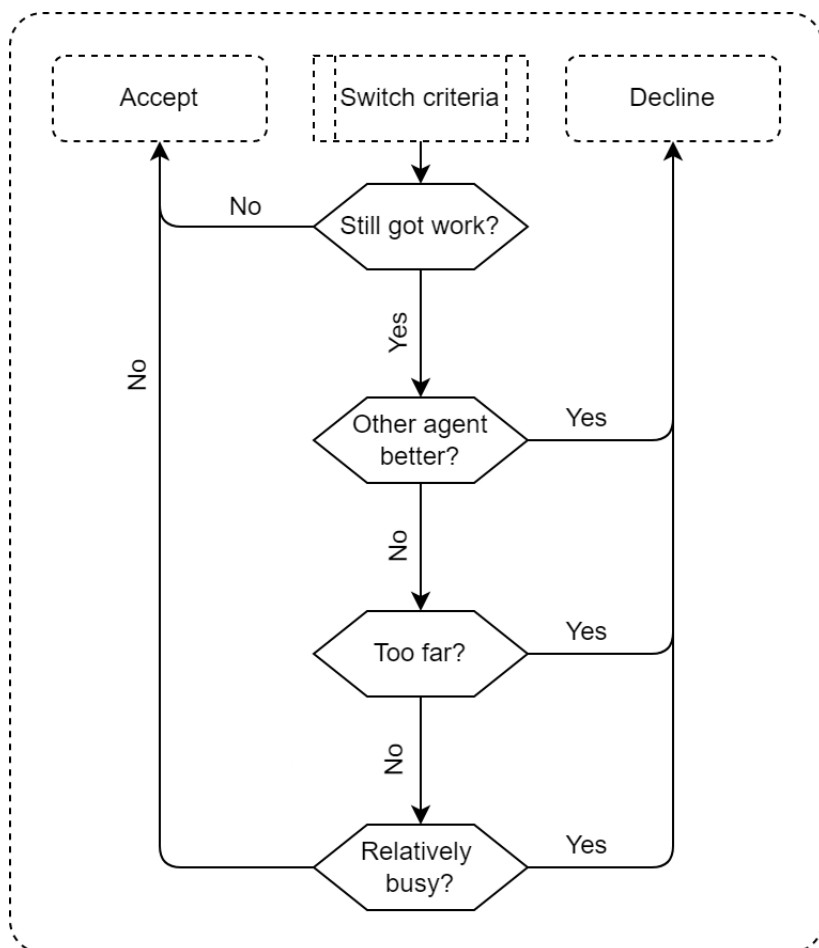

**Figure 3.** Selecting criteria for autonomous agents in the blanket search scenario.

The machine agent in the presented experiments has a strictly-defined knowledge representation. It holds for each agent an estimation of their skill levels for each type of skill. During the execution of the task, skill levels may be adapted based on interactions, for example a team member that refuses work. This knowledge representation works well when interacting with other robot agents, as these work with the same paradigm.

### 3.2. Urban Search and Rescue

Urban search and rescue (USAR) is a type of technical rescue operation that involves the localization, extrication, and initial medical stabilization of victims trapped in an urban area. An USAR mission is executed when there is a high chance of structural collapse due to, for example, natural disasters, war, or accidents.

Teams performing an USAR operation have to be well coordinated to make good use of each individual's skills. With the advancement in technology, robots were introduced in USAR operations [35]. Although currently these are mainly tele-operated, a lot of research is being completed in order to increase the autonomy of robots that assist in the USAR task [36,37]. This indicates the importance of team collaborations with autonomous robots and human team members in the future.

### 3.2.1. Objective and Tasks

Figure 4 shows an example of the simulated USAR-task that was created. The goal is to find victims in buildings and bring them to safety in the command post. Team members have to collaborate in order to clear rubble in front of a building, establish whether it is safe to enter a building, establish whether a victim needs treatment, treat victims, and carry them to the command post. The skills and skill levels of the team members differ significantly, depending on their role within the team. Only a rescue worker (human) can establish whether a victim needs first aid, and provide it when necessary, while only explorers (agents) are able to clear rubble in front of building entrances, and can enter collapsed buildings. Robot agents also have a battery that needs to be replaced by a human agent.

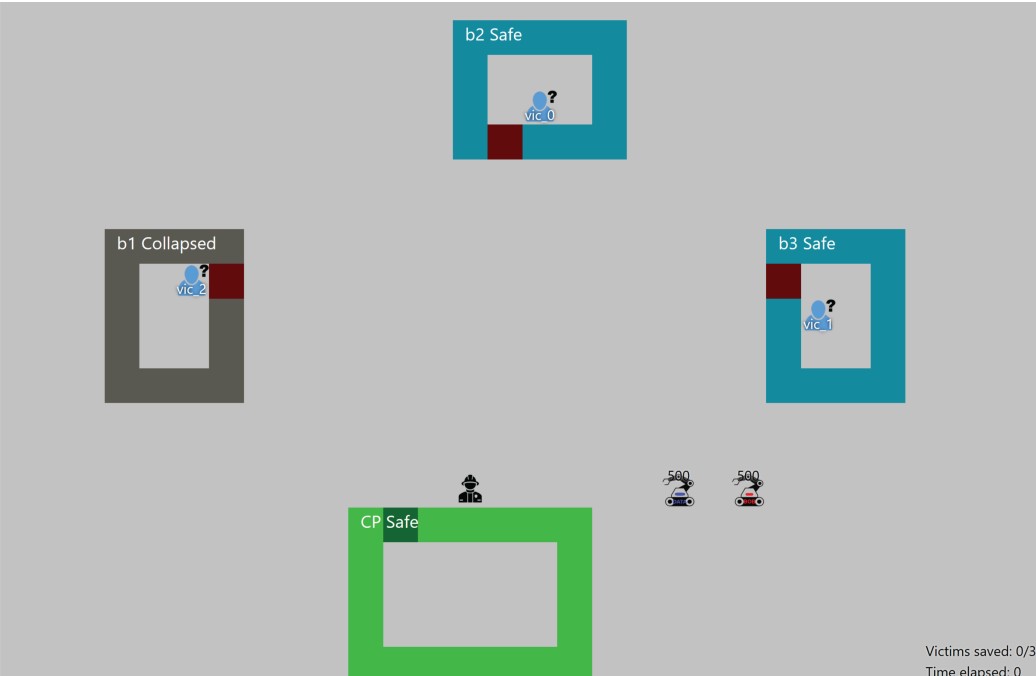

**Figure 4.** Screenshot of the simulated urban search and rescue (USAR) task. In this example, three team members collaborate to find and treat victims (represented by the blue person-like icons) in buildings that are either collapsed (grey walls) or safe to enter (blue walls). Red squares indicate rubble that needs to be cleared before the building can be entered. All victims need to be brought to the command post (building with green walls).

3.2.2. Team Members

In both the agent-only and human–agent experiment, AI agents fulfilled the role of explorer. Explorer agents differ in competency level between 0 and 1 on the tasks they are able to perform (determining the status of a building, clearing the entrance of a building, and transporting victims. In experiment 2, the human fulfilled the role of rescue worker, differing in skillset from the explorer agents.

The explorer robots exhibit a default behavior which is as follows:

- Travel to the closest building that has not been inspected yet;
- Inspect that building;
- Clear the entrance of rubble if there is any;
- Search the building for victims;
- Repeat 1–4 until all buildings are inspected and searched;
- Inspect closest victim until all victims are inspected;
- Bring closest victim to the command post.

From the default robot behavior may be deviated when other agents, human or robot, send requests for delegating tasks. Conducting actions takes time. How much time is required for a certain task depends on the skill level of the robot for that task. Figure 5 shows an overview of the behavior of a robot explorer in the USAR case. It shows both the behavior for the basic reasoner, as well as the mental reasoner. Please note that the mental model reasoner checks in between each task of the default behavior whether task delegation messages have to be sent, or whether task delegation messages were received.

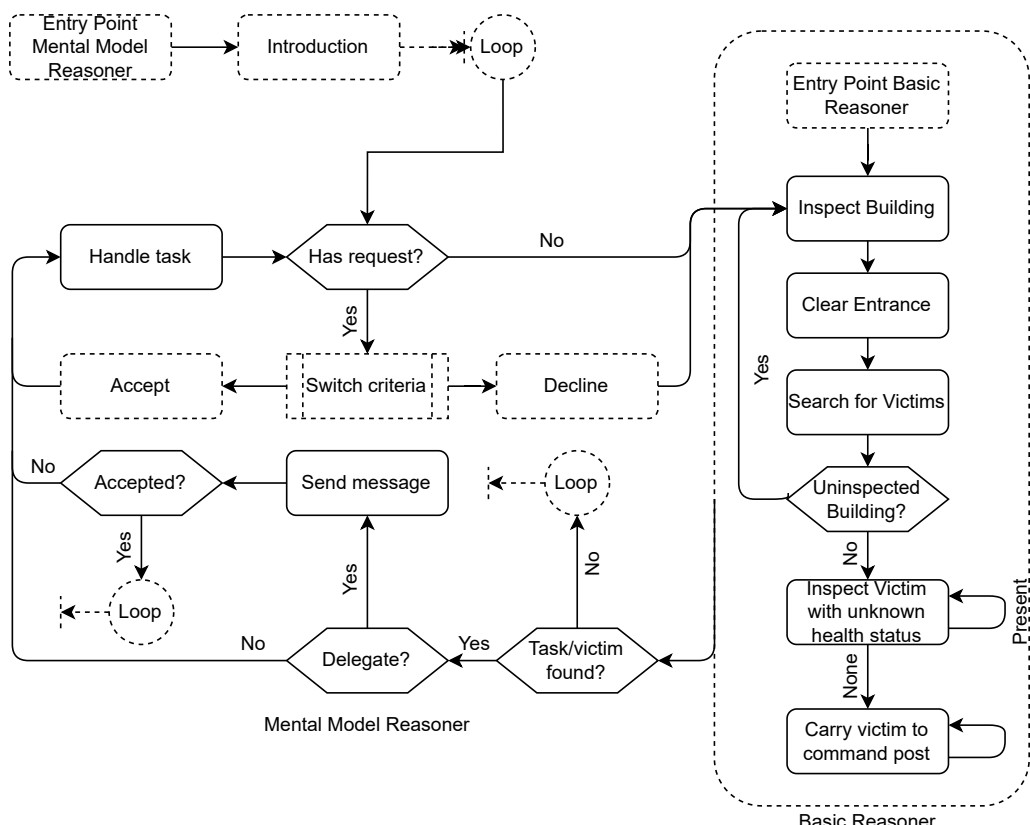

**Figure 5.** The basic reasoner behavior in the USAR case (**right**), and its mental model reasoner addition (**left**).

### 3.2.3. Agent Implementation

The agent implementation in the USAR task is similar to the one in the blanket search. Therefore only significant differences are mentioned in this section. All agents have a task model, containing information about the task context and the skill dependencies. The agents decide autonomously whether their own tasks should be delegated to someone else, and whether a request of another agent seems beneficial for the team. This is completed with task switch criteria. The task switch criteria are almost identical compared to the blanket search task, and agent again use a mental model based on holds skill-estimations for each agent for each type of skill. Some changes have been added to prevent deadlocks. The deadlocks can occur as in the USAR task agents might lack a skill completely, while in the blanket search tasks all agents could perform all tasks (although sometimes not efficient). The process of deciding whether or not to accept a task delegation is shown in Figure 6.

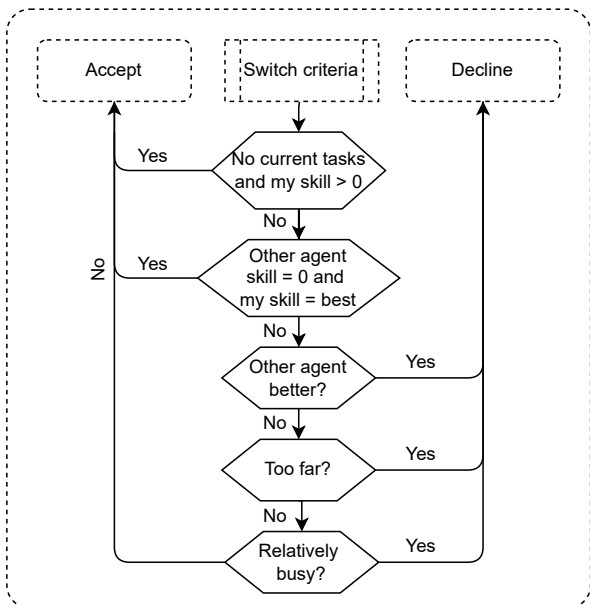

**Figure 6.** Task acceptance criteria for the USAR task.

## 4. The Value of Proactive Task Reassignment for the Team

The study concerns the question whether explanations of agents help a human to form an accurate mental model of the agent's capabilities. These explanations are provided within the team design pattern "proactive task reassignment" (PATRA), which is introduced in Section 4.1. If explanations do help, then humans should develop more accurate knowledge of the team members' competencies, and a better understanding of when to exchange tasks, and when not to. In turn, it is expected that the quality of collaboration, and the team's overall performance will improve. To show that accurate mental models indeed improve the team performance, we tested the value of accurate mental models in an team consisting of only agents for the blanket search use-case (see Section 3.1). These results are shown in Section 4.3.

### 4.1. Definition of TDP PATRA

Proactive task reassignment is defined in Table 1, according to the Team Design Pattern template as proposed by [9,10].

**Table 1.** Definition of TDP PATRA.

| Pattern Title | Proactive Task Reassignment (PATRA) |
|---|---|
| Behavior pattern | 1. The agent encounters a task. <br> 2. The agent deliberates whether delegating a certain task will be beneficial for the team performance. <br> 3. The agent decides whether or not to delegate the task and to whom. <br> 4. The agent sends a request to the other team member asking them to adopt the task. <br> 5. The team member receives the request. <br> 6. The team member considers whether adopting the requested task is beneficial for the team performance. <br> 7. The team member decides whether or not to adopt the task. <br> 8. Depending on the decision the team member either sends an accept message or a reject message. <br> 9. If the team member accepts the task, the requesting agent removes the task from their action list. If the team member rejects the task, the requesting agent keeps the task in its action list for the time being. |
| Positive effect | Team members can divide tasks based on skill and work pressure. There is a higher chance of tasks being performed by those team members that are most effective performing them. |
| Negative effect | One might potentially decrease overall team performance depending on the implementation of this pattern. For example, the mental models may be incorrect, or the exchange of tasks may not be beneficial for the total division of tasks among the team members. |
| Use when | The overall task performance depends on the division of tasks among the team members and can be improved by altering the division of tasks within the team, for example based on competencies or circumstantial possibilities for the different team members. |
| Example | A team of three actors works together in an urban search and rescue task. One of the actors is a medic, the second one is a fire fighter, and the third an ambulance driver. The medic searches for survivors in a room and runs into a fire hazard. He requests for the firefighter to come and extinguish the fire. The firefighter is more capable at extinguishing the fire, so the total team performance is expected to improve: the fire is put out sooner and more safely. |
| Design rationale | Members of well-functioning teams make use of each other's skills and expertise: assigning tasks to members with the required skill supports teamwork and team performance [38,39]. |
| Type | Individual |

### 4.2. Applicability of TDP PATRA for Hybrid Teams

In this paper, two use-cases are considered: a blanket search use-case and an urban search and rescue (USAR) use-case. These use-cases share the characteristic that their tasks are usually completed in a team setting and are quite complex and extensive. When working in a team, good coordination and making use of each other's skills is vital. The blanket search use-case might illustrate that even though all team members are able to perform all possible tasks, having a task division based on not only skills but also workload (and other environmental factors) can be beneficial for the team performance. The USAR use-case on the other hand, highlights that with a different division of skills, being able to reassign tasks based on workload and other environmental factors might also improve the overall performance.

The strength of TDP PATRA lies in the reassignment of tasks based on skills, workload, and other environmental factors. Creating an initial task division purely on skill works well in missions where all possible tasks are known beforehand. In the two use-cases described in this paper, this is not the case. In the blanket search use-case, the team members do not know yet what tasks they will find. For example, a fire might be present below deck,

but this is not yet known at the start of the blanket search. For the USAR use-case, while the locations of buildings may be known beforehand, the team members do not know the status of the buildings or the number of victims still inside. Due to this uncertainty in the world, it is important to be reactive to the world in the sense that team members can re-allocate tasks. Doing this purely based on skill might not be sufficient as utilizing less-skilled team members may improve the overall team performance as more work can be performed in parallel.

When considering TDP PATRA for other use-cases, it becomes clear that the TDP might especially support learning and collaboration in hybrid teams. It is especially important in hybrid teams to have a clear understanding of one's team members, as it is vital for team performance. A well performing team knows and trusts each other and communicates clearly and efficiently. TDP PATRA can help not only in finding the best team member for a task, but also in increasing understanding about why a team member wants to re-allocate a task. This might especially benefit hybrid teams, as it instructs the artificial team members to make their reasoning about their choice of task explicit to human partners.

TDP PATRA might especially be considered in situations where learning to collaborate with team members is one of the main goals, and where the team size is small (e.g., 2–8 team members). That is, the TDP is focused at supporting team members to learn about others and to use this information to divide tasks more effectively. In the long term, these learning outcomes facilitate understanding between team members, thereby improving effective collaboration of the team. In large teams, the TDP might not be able to benefit short-term task performance because of the significant communication effort that is required between all team members. In situations where there is less freedom for team members to devise tasks when collaborating (e.g., because the tasks and team roles are already clear prior to collaboration, or because there is a team leader that gives orders), PATRA might not be desired as it can impede progress by drawing team members' attention away from the task. Thus, PATRA mainly offers support in situations where it is beneficial to (re)distribute tasks when collaborating. For example, the TDP can support self-managing hybrid teams to learn about each others skills and preferences, to define roles within the team, and to establish a way of working for the task at hand.

When evaluating the added value of PATRA for a situation, team learning and performance on the short- and long-term basis needs to be taken into account. TDP PATRA is especially suitable if the expected (short- or long-term) learning outcome for the team is more important than the performance on the task at hand. In addition, it is also suitable for directly improving task performance, especially for tasks where expertise of team members has a large influence on the efficiency and/or effectively with which a task is performed.

### 4.3. The Value of Proactive Task Reassignment

This section presents an experiment into the effects of the TDP PATRA when all agent team members have a complete and accurate mental model of each other. The purpose of this experiment is to investigate whether the pattern improves team performance in the selected use case, when compared to teams that do not exchange tasks. Testing the effectiveness of the TDP is important, as in the main study (Section 5) we investigate whether adding the element of "providing explanations" to the TDP leads to a better understanding in the human agent, and whether it yields additional effects on the team's functioning and performance.

A team of five artificial agents performed the blanket search as efficiently as possible. The task environment distinguished five routes, with each route being assigned to an artificial agent. Agents differed in skill level on handling the various malfunctions, but not on solving incidents. One agent had equally moderate skills on all malfunctions; the other four agents were characterized by having high skill on repairing a particular malfunction, and below-average skill on the other malfunctions.

Two types of scenarios were administered: (1) An easy scenario, in which the probability of incidents and malfunctions was low. Incidents were uniformly distributed over

the compartments of the ship; (2) A difficult scenario, in which there is a high likelihood of fires near mobility systems; victims near SeWaCo systems; and malfunctions in energy near SeWaCo systems.

Three types of team agents were developed: (1) "BR" indicates that the team consists of agents with a basic reasoner. Agents of BR-teams performed the tasks in prefixed division without exchanging tasks; (2) "SBR" indicates that the team consists of agents that have a skill-based reasoner, enabling them to take knowledge about the skill levels of all agents into account when requesting task exchanges; (3) "SCBR" indicates that the team consists of agents that have a skills and circumstances reasoner, enabling them to take skill levels of all agents into account, as well as distance and workload.

We measured: (1) how often tasks were exchanged; (2) the time that agents are idling; (3) how skilled agents are on average on the tasks that they perform; and (4) the total time needed to complete the blanket search. Each team performed 100 blanket searches on the easy and difficult scenario, and the measurements were averaged over all 100 runs. The results are presented in Table 2.

**Table 2.** Testing the value of proactive task reassignment with perfect mental models.

| | BR | | SBR | | SCBR | |
|---|---|---|---|---|---|---|
| | Easy | Difficult | Easy | Difficult | Easy | Difficult |
| Task Exchange Percentage | 0% | 0% | 28.5% (28.5%) | 24.7% (24.7%) | 25.4% (28.1%) | 19.9% (23.5%) |
| Idling Percentage | 65.2% | 54.0% | 44.9% | 43.8% | 47.5% | 48.4% |
| Task Competency | 0.58 ±0.10 | 0.60 ±0.08 | 0.72 ±0.10 | 0.72 ±0.07 | 0.70 ±0.10 | 0.71 ±0.07 |
| Completion Time | 2742 ±972 | 3453 ±1141 | 1468 ±511 | 1982 ±650 | 1596 ±667 | 2289 ±952 |

For the task exchanges, the number in the brackets in the first row of Table 2 indicates how may requests were made on average, and the number outside the brackets the average number of accepted requests. The BR team obviously sends no requests as they follow a fixed task division. We see that approximately a quarter of all tasks are exchanged. In the difficult scenario this percentage is for SCBR somewhat lower as there are many tasks and workload is taken into account. We also see that SCBR does not grant all the requests. This is because they consider its location and workload upon receiving a request. These factors are not known to the requesting agent and may lead to rejected requests.

Once all tasks of an agent are performed, an agent falls idle. As Table 2 shows, the BR team spend relatively more time being idle than agents in the other two type of teams. Thus, re-assignment of tasks, as induced by the PATRA Team Design Pattern, supports the teams to deploy their agent capacity more efficiently. The SCBR has a slightly higher idle time. This might be caused due to the slightly higher rejection level of requests, for example, if a task is too far away according to the reasoning criteria, the SCBR agent does not accept the task reassignment.

Table 2 also shows that PATRA increases the average task-specific competency of an agent that is performing a task. On average agents that use PATRA solve tasks that are better suited to their capability.

The measure that reflects overall performance of a team is the time to complete the entire blanket search. Table 2 that teams of agents that apply the TDP PATRA complete the blanket search faster than teams who do not. Interestingly the SCBR agents required more time to complete the blanket search task than the SBR agents. It was hypothesized that the SCBR team would be the quickest, as these agents take into account the when to accept or reject a task take-over request. It was believed that this would prevent them from accepting requests that increase unbalance in the agents' workloads or involve long-distance traveling. It may be that the chosen thresholds for workload and distance are not optimal for the

scenario. The time gained by solving problems that match a high agent skill currently outweighs the additional overhead.

In general, we can conclude that TDP PATRA improves the team performance when it is being used with an accurate mental model. It is known that dynamic allocation of tasks to team members is one reason for expert human teams being so effective and efficient [40,41]. It is, therefore, of vital importance to form an accurate mental model to utilize the team design pattern efficiently. How artificial team members can support the human in forming an accurate mental model by providing explanations is discussed in the next sections.

## 5. The Effects of Explanations on Human Mental-Model Shaping and on Team Performance

This section describes the setup of the experiments of measuring the model that a human forms of artificial team members within a human–AI team and the implications on team performance. During the execution of the team task, the human creates a mental model of the task, its team members, and him/herself. As a human is not bound by written computer code, any aspect may be included in the mental model. Even aspects that are not relevant in the implementation may appear relevant to the human until enough empirical evidence suggests otherwise. It is, therefore, interesting to measure the effect of additional explanations of the agents behavior on how the mental model of the human is shaped.

An ideal robot agent would not stick purely to its own mental model paradigm, but would try to understand the mental model that the human is currently using in order to improve the team performance. This is a mental model of another agent's mental model, so to speak. Such a meta mental model is out of scope of this research, and robot agents stick to their own mental model paradigm. This section purely concerns the human side of mental model shaping, and how explanations may help this process.

The experiments presented in this section measure how the human mental model is being formed during the execution of the blanket search and urban search and rescue team tasks. More specifically, the effects of providing additional explanations during the task delegation interaction with team members is investigated. The hypothesis is that by providing explanations, the human mental model will more quickly converge to an accurate representation of the team task and the skills levels of the agents. In order to test this hypothesis, a controlled experiment with humans was conducted. The experimental setup is discussed in Section 5.1, and Section 5.2 presents the results of this experiment.

### 5.1. Method

This section describes the experimental setup for measuring whether explanations during task reassignments indeed enhance the quality of the mental models of human team members.

#### 5.1.1. Participants

A total of 44 subjects participated in this experiment. Data of eight participants was left out of the analysis due to technical issues early on in the experiment. The experiment was approved by the TNO ethics committee. Inclusion criteria were an academic background and experience in the healthcare domain. All experts stated to have sufficient technical ability to participate in an experiment held in a digital environment. Compensation and travel reimbursement of EUR 25 was offered.

#### 5.1.2. Design

The goal of the experiment is to investigate whether explanations, that agents either provide or do not provide in task exchange requests, have an effect of the quality of the human's mental model. This is being investigated using a within-subjects design, in the context of two different task domains: blanket search, and urban search and rescue. Each participant participates in both tasks, with either an explaining agent as team partner, or

with a non-explaining agents as team partner. The order in which the tasks were being administered was balanced. The design can be seen in Table 3.

**Table 3.** Within-subjects design of experiment into the effects of explanation in human–agent communication about task exchanges. Effects of explanation were measured on human understanding (mental model), team functioning (within-team collaboration), and on team performance. Two task were used: blanket search (BS) and urban search and rescue (USAR). Order of tasks, and explanation condition were balanced.

| Cohort | N | First Task | | Second Task | |
|---|---|---|---|---|---|
| 1 | 17 | BS | (explaining agents) | USAR | (non-explaining agents) |
| 2 | 19 | USAR | (explaining agents) | BS | (non-explaining agents) |
| 3 | 19 | BS | (non-explaining agents) | USAR | (explaining agents) |
| 4 | 17 | USAR | (non-explaining agents) | BS | (explaining agents) |

Measures of understanding (quality of the human's mental model); of team functioning (quality and fluency of human–agent collaboration), and team performance are being collected after and during each run.

### 5.1.3. Materials

A simulation of the USAR-task was developed using the Python programming language and a software library called MATRX (human–agent teaming rapid experimentation)[1]. MATRX allows rapid design, simulation, and testing of 2D top–down, grid-based environments in which multiple agents can perform tasks collaboratively. Moreover, it contains a chat window as part of its interface, allowing agents to communicate. The behavior of the agents can be programmed or controlled by a human player. Both the blanket search (Section 3.1) and the urban search and rescue use case (Section 3.2) were implemented in the MATRX framework.

In the experiment, participants were asked to perform each team task (Blanket search and USAR) in four different scenarios. The scenarios were increased in complexity in terms of the number of subtasks that needed to be executed in order to complete the overall team task. For the blanket search task, the scenarios differed in the number of malfunctions and incidents. In the USAR task, complexity was increased by adding more victims and more collapsed buildings. By gradually increasing the complexity the participants were eased into the more difficult situations without overwhelming them. In all scenarios, the skillset of the agents remains constant.

### 5.1.4. Agent Skill Levels

Table 4 shows skill levels of team members per incident and malfunction in the blanket search. A 1 indicates skilled; 0.1 indicates a low proficiency. All team members are equally skilled to solve incidents, and they are instructed and programmed to solve found incidents immediately. Team members differ, however, with respect to their skill in repairing malfunctions. All members are able to eventually repair any malfunction, but members with a high skill level require substantial less time for repair.

**Table 4.** Skill levels assigned to the members of the blanket search team.

| Skill | Agent: SKY | Agent: HAL | HUMAN |
|---|---|---|---|
| SEWACO | 1 | 0.1 | 0.1 |
| MOBILITY | 0.1 | 1 | 0.1 |
| C4I | 1 | 0.1 | 1 |
| ENERGY | 0.1 | 1 | 1 |
| FIRE | 0.5 | 0.5 | 0.5 |
| LEAK | 0.5 | 0.5 | 0.5 |
| ASSIST | 1 | 1 | 1 |

Table 5 shows how skilled the agent team members in the USAR scenario are. The skills were defined in such a way that the agents are specialized in certain tasks. Please note that some tasks are require specific team members.

**Table 5.** Division of skills of the two artificial team members for the USAR task.

| Skill | Agent: BOB | Agent: DATA | HUMAN |
|---|---|---|---|
| DETERMINE VICTIM'S HEALTH | 0.7 | 0.1 | 1 |
| CARRY VICTIM | 0.5 | 1 | 1 |
| DETERMINE BUILDING STATUS | 1 | 0.5 | 0 |
| CLEAR ENTRANCE | 0.1 | 1 | 0 |
| TREAT VICTIM | 0 | 0 | 1 |

### 5.1.5. Procedure

It takes about 90–120 min for a participant to participate in the experiment. First, the participants are given information about the purpose of the study and about the tasks that they will perform in collaboration with the software robot-agents.

Figure 7 gives an overview of the experiment. First, participants were welcomed, received general instructions, and were asked to sign an informed consent form. Then, two blocks of tasks were executed, one with all the blanket search scenarios, and another for all the USAR scenarios. The order in which the participants did these two tasks was balanced over the participants. One task block contained instructions of the task and a practice run. Then, the four scenarios of each task were executed (see Section 5.1.2). After all scenarios of the same use case were completed, the participants were asked to fill in a question form. This process is shown in Figure 7. The measures, as described in Section 5.1.6, are recorded during the task runs.

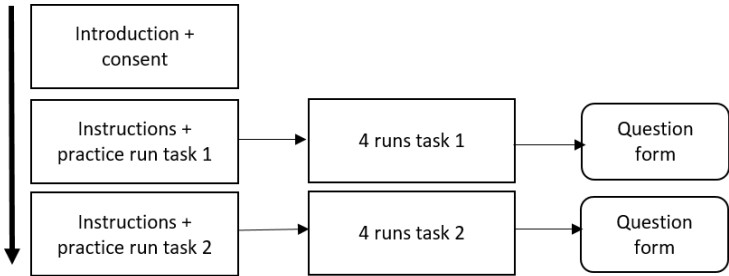

**Figure 7.** The flow of the experiment.

### 5.1.6. Measures

In order to investigate the effect of agent explanations on human mental model formation and team performance in a human–AI team, several aspects were measured during the task, and in a questionnaire after completion of each task. Table 6 shows all measurements that were performed in the experiment. Team performance was objectively measured by logging quantitative aspects during task execution, and subjectively by evaluative questions in the questionnaire (e.g., the Collaboration fluency questionnaire from Hoffman [42]). To externalize the mental model that was created by the participants during the experiment, several questions were added to the questionnaire about the reasoning of participants and the reasoning of other team members during the task. For the latter type of questions, participants were presented with visualizations of two scenarios in the BS and USAR task, and were asked about the expected behavior of a particular agent in the scenario (e.g., "In this scenario, will agent HAL accept the request from agent SKY? Why (not)?").

**Table 6.** Overview of the measures that were recorded during task execution. TP = measurement of team performance, MM = measurement of mental model.

| Measurement | Method (Range) | Description |
|---|---|---|
| Task request behavior (MM) | Q item | Four open questions on the reasoning of participants when sending task request, or responding to task requests from the robot agents. |
| Best agent for task score (MM) | Q item (0–1) | Ratio of correct answers on questions asking which team member would be best suited for a specific subtask. Participants also indicated the certainty of their answers on these questions on a 1–5 Likert scale. |
| Best task for agent score (MM) | Q item (0–1) | Ratio of correct answers on questions asking which subtask would be best suited for a specific agent. Participants also indicated the certainty of their answers on these questions on a 1–5 Likert scale. |
| Agent behavior prediction score (MM) | Q item (0–2) | Mean score on questions about the expected request, accept or decline behavior of a particular agent for particular scenarios. Participants also indicated the certainty of their answers on these questions on a 1–5 Likert scale. |
| Collaboration fluency (TP) | Q item (1–5) | Mean score on the Collaboration Fluency questionnaire [42]. |
| Explanation evaluation (TP) | Q item (1–5) | Mean score on two evaluative questions about the explanation provided by the robots. |
| Mental effort (TP) | Q item (1–20) | Mean score on the Rating Scale Mental Effort (RSME) [43]. |

5.1.7. Analyses

We performed multiple linear mixed effects analyses in order to investigate the effect of agent type (explaining vs. not explaining), task type (blanket search vs. urban search and rescue), and their interaction on the measurements in Table 6. This type of statistical analysis subtracts the random variability within each participant's outcomes, thereby correcting for potential effects of the order in which participants carried out the tasks while collaborating with a particular type of robot agent (see Table 3).

Responses on the open questions about task request behavior (i.e., reasons to request a robot-agent to take over a task, and reasons to respond to task requests from robot-agents) were analyzed by two authors. We independently coded and categorized responses from five randomly chosen participants. We then discussed our results and developed a closed coding scheme for further analysis. Keywords were assigned to each response, and we calculated the sum of their occurrence for each question. Answers on the situational judgment questions in which participants predicted agents' task request behaviors were scored based on their accuracy (i.e., answer on the yes/no question whether an agent would reassign/accept/reject a particular task), and the number of correct arguments they include in their answer (e.g., 'HAL is less skilled than SKY'/'HAL is too busy'). With each question, a total score of 4 points could be obtained (1 point for accuracy, and 1 point for each correct argument). The scores on the questions concerning the best agent for a task, best task for an agent, and the predicted agent behavior with respect to task requests were normalized between −1 (all questions answered incorrectly) and 1 (all questions answered correctly) prior to analysis.

*5.2. Results*

The results of the experiment are presented in two sections. In Section 5.2.1, we show the findings regarding participants' mental model of robot team members. Section 5.2.2 shows the findings regarding the team performance. For both the mental model and performance results, we investigated differences between participant groups working with robot agents that provided explanations, and with agents that did not. We also compare results between the blanket search and urban search and rescue task.

Prior to analysis, we checked each variable for any outliers and for the presence of other effects. Although some outliers were found, they were relatively small ($<2.5 \times$ SD) and we judged them be representative of the population, so we decided to keep them in the analysis. We found one small interaction effect between order and task for the agent behavior prediction score, which is discussed in Section 5.2.1. Table 7 shows the results of all statistical analysis that were performed on the quantitative data, which will be further discussed in Sections 5.2.1 and 5.2.2.

**Table 7.** Results of the linear mixed effects analyses on all measurement variables. Results in bold text indicate statistical significance at $\alpha = 0.05$.

| Variable | Main Effect of Explanation Condition (Explanation vs. No-Explanation) | Main Effect of Task Type (BS vs. USAR) | Interaction Effect (Expl. Condition $\times$ Task Type) |
|---|---|---|---|
| Best agent for task score | $t(69) = 1.22, p = 0.223$ | **$t(69) = 3.76, p < 0.001$** | $t(36) = 1.22, p = 0.223$ |
| Certainty of best agent answers | $t(64) = 1.91, p = 0.061$ | $t(64) = -0.72, p < 0.471$ | $t(36) = 0.46, p = 0.649$ |
| Best task for agent score | $t(72) = 0.22, p = 0.824$ | $t(72) = 1.51, p < 0.136$ | $t(36) = 0.18, p = 0.859$ |
| Agent behavior prediction score | **$t(70) = -2.21, p = 0.030$** | $t(70) = -1.12, p < 0.267$ | $t(36) = -0.29, p = 0.776$ |
| Certainty of agent behavior predictions | $t(72) = -0.46, p = 0.065$ | $t(64) = -1.02, p < 0.310$ | $t(36) = 1.90, p = 0.066$ |
| Collaboration fluency | $t(68) = -0.18, p = 0.086$ | $t(68) = 1.74, p < 0.087$ | $t(36) = 0.52, p = 0.601$ |
| Explanation evaluation | n.a. | $t(24) = -1.28, p < 0.214$ | n.a. |
| Mental effort | **$t(69) = 2.58, p = 0.012$** | **$t(69) = 3.51, p < 0.001$** | $t(36) = -1.73, p = 0.093$ |

### 5.2.1. Results on the Development of Mental Models

In order to investigate whether people managed to develop accurate mental models of their robot team members, we analyze their ability to choose the best agent for a specific task (**best agent for task score**), to choose the best task for a specific agent (**best task for agent score**), and to predict the task request/response behavior of both agent team members in specific scenarios (**agent behavior prediction score**). Moreover, we take into account the results on the open questions about task request and response behavior.

Figure 8 shows the aggregated results of the questions in which participants indicated the **best agent for a task** in either the BS or USAR scenario. Overall, scores indicate that, on average, participants perform much better than chance level (which corresponds to a score ratio of 0 due to our applied scoring mechanic). A similar result is found for the reversed version of this question (**best task for agent score**), for which an average of 56% of the maximum score was achieved for BS, and 40% for USAR. We found no main effects of explanation condition on these scores. For **best agent for task type score**, we found a small effect of task type, in which participants were able to better identify the best agent for a task in the USAR scenario compared to the BS scenario (see Table 7). Although it looks like there is a trend in which the best agent scores are higher in groups where participants collaborated with agents that did not provide explanations when sending or responding to task requests, this results is not statistically significant (see Table 7). Finally, we analyzed the certainty with of participants in choosing the best agent for a given a task (**Certainty of best agent answers**). On average, participants were quite certain that the agent they chose was the best choice for a task (mean certainty score of 3.79 out a 5-point Likert scale), and this did not significantly differ between Explanation and Task conditions.

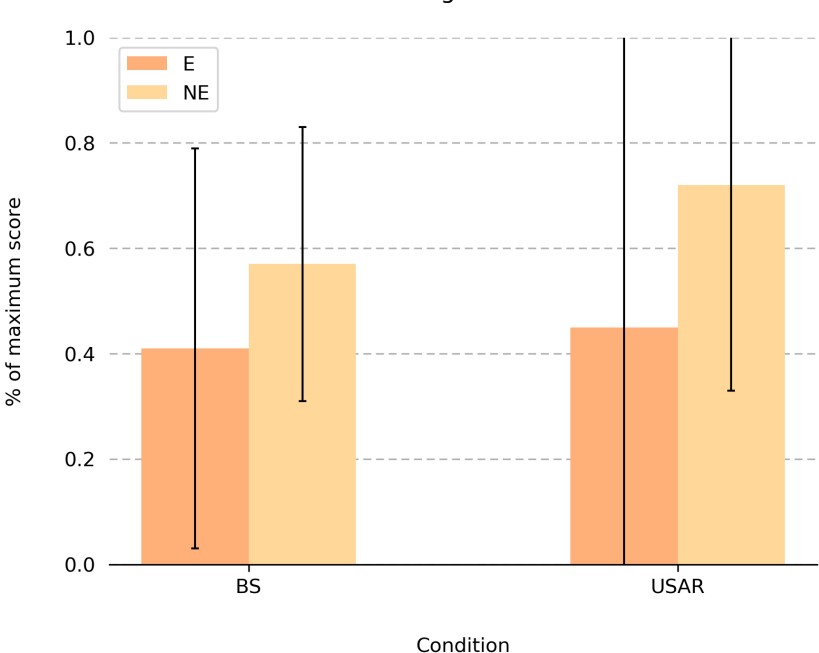

**Figure 8.** Means and SDs for the score ratio of participants when choosing the best agent for each task in the blanket search (BS) and urban search and rescue (USAR) use-cases. Ratios are compared between groups working with explaining (E) agents and working with non-explaining (NE) agents. The main effect of task (BS vs. USAR) is significant at $\alpha = 0.05$.

Figure 9 shows the mean score ratios for the questions in which participants predict the task request and response behavior of the robot agent team partners. On average, scores were high (74% of total score for the BS task, and 60% for the USAR task), indicating that participants were quite good at predicting when an agent would request a team member to take over a task, and how this agent would reply to task requests of others. Although the difference between task type was not statistically significant, there is a significant effect of explanation condition on the **agent behavior prediction score**. That is, the prediction scores were significantly higher after collaborating with agents that explained their task request/respond behavior (M = 0.74, SD = 0.28) than after collaborating with agents that did not provide explanations (M = 0.59, SD = 0.25). This same effect was found within the BS and USAR case (i.e., there was no interaction effect between explanation condition and task type). Interestingly, participants reported a high **certainty of agent behavior predictions** (average of 3.44 out of a 5-point Likert scale), and these certainty scores did not differ between explanation conditions, or between task types. We however found a small, but statistically significant effect between order and explanation condition for the certainty ratings on the behavior prediction questions (**Certainty of agent behavior predictions**): for the BS task, on average, people were more certain of their behavior predictions for NE-agents when they first collaborated with E-agents. For the USAR task, this effect was reversed; on average, people were more certain of their behavior prediction for E-agents when they first collaborated with NE-agents.

The improved ability to predict agents' task request/respond behavior (**agent behavior prediction score**) when receiving explanations from those agents was also supported by the answers on the open questions about this behavior. After collaborating with explaining agents, people mentioned more valid arguments (i.e., containing task/mental load, distance to a task, and/or skills) for sending and receiving task requests to and from others (an average of 7.67 valid arguments per participant) than after collaborating with non-explaining agents (an average of 4.67 valid arguments). Within those arguments, workload is mentioned much more frequently by participants after working with explaining

agents (average of 1.00 per participant), than after working with non-explaining agents (average of 0.47 per participant). Moreover, after completing the BS task, people more often mention skill differences between agents, and a task list that is contained by agents in one of their answers, than after completing the USAR task (on average 2.37 vs. 1.61).

**Figure 9.** Means and SDs for the score ratio of participants when predicting the task request and -respond behavior of agents for particular scenarios within the blanket search (BS) and urban search and rescue (USAR) use-cases. Ratios are compared between groups working with explaining (E) agents and working with non-explaining (NE) agents.

### 5.2.2. Results on the Effect on the Subjective Team Performance

On average, participants rated the experienced collaboration as rather fluent in both tasks and explanation conditions, as indicated by their general agreement with most 5-point Likert scale items of the **collaboration fluency** questionnaire (M = 3.15, SD = 0.43). The ability of agents to provide explanations when sending and receiving task requests did not have an effect on the fluency ratings, nor did the ratings differ between task types. However, on average, the agents' explanations were deemed valuable by participants (**explanation evaluation**) as indicated by their level of agreement with the two 5-point Likert scale items about the explanations (M = 3.66, SD = 0.67). This rating did not significantly differ between BS and USAR task types. Lastly, we analyzed the results on the rating scale for **mental effort**. Performing the USAR task was rated as requiring significantly more mental effort (M = 12.11, SD = 2.37) than performing the blanket search task (M = 10.44, SD = 2.08). No statistically significant effect of collaborating with either explaining or non-explaining agents was found, although a trend was visible in which the mental effort was rated as being lower when collaborating with explaining agents (M = 10.96, SD = 2.12) as compared to non-explaining agents (M = 11.59, SD = 2.33).

### 6. Discussion

In hybrid human–agent teams, people and artificially intelligent agents collaborate to achieve a common team goal. A way to do this effectively is by making use of Team Design Patterns (TDP) [10], which are formalized, proven solutions for effective teamwork. This paper investigated the effects of the TDP "proactive task reassignment" (PATRA) on the functioning and performance of a human–agent team. The concept behind PATRA is regulating task delegation within the team based upon the competencies of team partners and their current workload, while controlling for overhead due to task exchanges to uphold team efficiency. A pilot study demonstrated that PATRA improves performance of agent–

agent teams. Earlier research has shown that providing explanations to experiences are important for collaboration and for learning (e.g., [16,29,44]. This inspired the main study of this paper, investigating the effects of explanations during PATRA on the collaboration and performance of hybrid human–agent teams. Furthermore, it is tested whether explanations support the humans to develop mental models of their team partners and of the team.

### 6.1. Discussion on the Effects of Explanations

Intuitively, assigning tasks to the most competent team member is more effective than when each member simply executes the tasks that happen to be on its route. Exchanging tasks can especially be beneficial if the team consists of members with each having their own speciality concerning the various tasks. However, exchanging tasks is only profitable for the team if the benefits exceed the costs of deliberating whether or not to exchange a task, and of the costs involved in executing task exchange (e.g., movement, work load disturbance). In order to make good assessments concerning task exchange, team members need to have adequate and dynamically updated internal representations of task, team, and context. The development and maintenance of mental models is, therefore, of critical importance. In the main study of this paper, it was investigated whether explanations during the execution of PATRA had an effect on the quality of the human team member's mental model, and on the team's performance.

Results showed that explanations supported the human's understanding of the conditions under which task exchange is profitable, and the awareness of the competencies of team members. This finding was more profound for the USAR-task than for the BS-task. The USAR-task involved hard dependencies, thus demanding team members to organize coordinated action. This feature probably incited humans to become well-informed about the qualities of their team partners. The blanket search task did not involve such hard dependencies. Although exchanging tasks with team members would improve performance, the BS-task can also be completed without. In principle, the human would be able to complete all of its tasks, even though for some tasks with rudimentary skill only. The fact that task exchange in the BS-task is not imperative may have hampered humans to learn about the teammates' competencies.

Explanations did not affect the human's experience regarding the fluency of collaboration. Participants' appreciation of the collaboration with team agents was higher for the USAR-task than for the BS-task. We think that this outcome is caused by the nature of the USAR-task. As this task involves hard dependencies, it inevitably demands coordinated action between the human and the other agents in the team, which likely resulted in a higher appreciation.

The here presented research addressed an important challenge of human–AI co-learning [1,14], namely that team members need mental models of one another's capabilities. Such capabilities are essential in situations where the cost-of-failure is high [45]. We believe that PATRA is one of many team design patterns that potentially improve the coordination of human–machine teams [46], in which theory of mind plays an important role.

### 6.2. Limitations

The aim of this study is to investigate the effects of team design patterns and the effects of explanations in the TDP on human understanding of task and team. To address these questions, we designed a task, a team, and designed the experiment. All of these may have imposed limitations which we will discuss below.

First, we consider the selected task. There are not yet many real-life situations in which humans and artificially intelligent agents work together as a team on a task. Following other researchers [47], we, therefore, decided to construct a simulation of tasks with representative features of the respective domains (i.e., battle damage repair, and urban search and rescue). However, we realize that results obtained within these experimental task cannot automatically be transferred to the associated tasks in real life. Similarly, it is uncertain whether the effects of TDPs can be generalized to other domains (also see

Section 4.2 for a discussion on applicability of the TDP). Further research is needed to validate such claims.

Secondly, we consider the sample of participants. The participants of this research were recruited among the general public, with the selection criteria of having an academic background and having no experience in the selected domains. When for a real-life application humans are being teamed with agents, the humans are likely to have domain experience. For example, a navy sailor is likely to bring in its mental model of the task and environment when he or she starts working with an agent partner. This may affect the nature of teaming, and may thus yield different outcomes than shown in the present study.

Thirdly, we consider the developed experimental tasks. At all times throughout the experiment, participants were able to perceive the other agent team members at work. This was to ensure that participants could execute the TDP as intended by its design (e.g., by including the positions of all other team members). Thus, the results are obtained in a setting that is more transparent than situations will be in real life, as in real life it is not always possible to maintain fully informed about team members.

Finally, we consider the experimental design. For practical reasons we adopted a within-subjects design. This implies that each participants performed the BS and USAR task once, in either the explanation or the non-explanation condition (see Table 3). As a result, we used a multiple linear mixed effects to analyze effects. A more straightforward approach is to use a between-subjects design. This would reduce the time-on-task for participants, but would require significantly more participants.

*6.3. Future Work*

The current research presented the value of explanations in a single team design pattern. The question may be asked what the influence of team design pattern characteristics are on the shaping of the mental model of the human.

In contrast to the focus of the present paper on manipulations (i.e., PATRA, and explanations) intended to improve the human's mental model of the team agents, future work includes investigating how artificial agents develop a mental model of the human partner.

Another line of future research is to develop a communication model that enables humans to share their knowledge and understanding with collaborating artificial agents, and vice versa. The model needs to be able to accommodate the intrinsically different nature of representations that humans and agents have of their worlds. Such a communication model will be of benefit for implementing explainability of team actions and reasoning [48].

**Author Contributions:** M.P.D.S., T.A.J.S., K.v.d.B., O.H.V., T.H. and K.H.J.V. contributed equally. All authors have read and agreed to the published version of the manuscript.

**Funding:** This research was funded by the Dutch Ministry of Defense under grant number 060.38570.

**Institutional Review Board Statement:** The study was conducted according to the guidelines of the Declaration of Helsinki, and approved by the Institutional Review Board (or Ethics Committee) of TNO (Toetsingscommissie NWMO TNO, approved on 31 August 2020).

**Informed Consent Statement:** Informed consent was obtained from all subjects involved in the study.

**Data Availability Statement:** The code that let participants execute the scenarios, the automatic measures of the system during the scenario, as well as the raw data of the questionnaire answers have been made available to the public (https://doi.org/10.17632/k4yh2gf2xw.1).

**Acknowledgments:** We would like to acknowledge Jasper van der Waa, Jurriaan van Diggelen, and Marieke Peeters for their support during this study.

**Conflicts of Interest:** The authors declare no conflict of interest. The funders had no role in the design of the study; in the collection, analyses, or interpretation of data; in the writing of the manuscript, or in the decision to publish the results.

## Notes

1    https://matrx-software.com/.

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
