# Peer review of "“I’m Afraid I Can’t Do That, Dave”; Getting to Know Your Buddies in a Human–Agent Team"

_systems, doi:10.3390/systems10010015_

Round 1
Reviewer 1 Report
The present manuscript analyses the challenges that face hybrid teams and supports the design of effective human-agent collaboration.
This paper introduce a Team Design Patterns (TDP) called “proactive task reassignment” (PATRA) to assign tasks more effectively among team members by utilizing the different strengths and weaknesses of humans and agents.
Major issues:
- Conclusions should be improved to underline the differences between the old anf the new proposed method
- Experimental part should be improved
- Should be created a literature review section
Minor issues:
- Moderate English changes required
- Need a moderate layout restailing
Author Response
We appreciate the time and effort that you have dedicated to providing your valuable feedback on our manuscript. We are grateful for your insightful comments on our paper. We have been able to incorporate changes to reflect most of the suggestions. We have highlighted the changes within the manuscript.
Here is a point-by-point response to your comments and concerns.
Comment 1: Conclusions should be improved to underline the differences between the old and the new proposed method
Response: Thank you for pointing this out. We agree with this comment. Therefore, we have restructured the conclusions section and specifically pointed out differences between the new and old method.
Comment 2: Experimental part should be improved
Response: Thank you for this suggestion. We have considered adding a comparison to other team-design patterns and the application to other use cases. We however believe that the value of explanations, which is the focus of the article, is sufficiently demonstrated within the current team-design pattern and that the use cases are suitable environments for the experiments.
Comment 3: Should be created a literature review section
Response: Agree. We have, accordingly, revised the introduction and created a literature section.
Comment 4: Moderate English changes required
We have proof-read the entire article and improved several language issues.
Comment 5: Need a moderate layout restyling
We have double-checked the tables and figures for styling issues.
Reviewer 2 Report
The authors propose a decentralized task delegation strategy called PATRA. They performed a comprehensive evaluation of the different characteristics present in the design.
Unfortunately, I am missing a comparison with other strategies to solve the same problems. Thus, the authors fail to answer whether PATRA makes sense as a strategy, especially for the proposed use cases. For example, what is the team size for PATRA to have a significant impact? A centralized coordinator/dispatcher might allocate resources/agents better in an extensive search and rescue operation, while PATRA might be better for small teams. In addition, when we think about military strategies, for example, there is the idea of a squad leader, a person in the unit that has the best overview of the capabilities of each team member and usually has excellent situational and strategical awareness. So how does PATRA stack against the team leader model? Even when considering hybrid teams, the team leader model unburdens each agent from being fully aware of the other's capabilities. Furthermore, how is PATRA different from on the field problem-solving?
Regarding the choice of use cases, I'm not sure the component of explaining your capabilities in a critical situation is an adequate strategy. I am not sure there is time to update capability models in the field. You either can handle the situation or report it and move on. Therefore, I would encourage the authors to find use cases where PATRA makes sense. Furthermore, with the proposed use cases, the fact that the paper addresses hybrid teams is irrelevant. PATRA is a strategy for resource allocation in a problem-solving scenario. Whether the agent is a person or a robot becomes irrelevant.
Finally, I would ask the authors to revise the paper to write more succinctly. Please compress the introduction and the discussion.
Author Response
We appreciate the time and effort that you have dedicated to providing your valuable feedback on our manuscript. We are grateful for your insightful comments on our paper. We have been able to incorporate changes to reflect most of the suggestions. We have highlighted the changes within the manuscript.
Here is a point-by-point response to your comments and concerns.
Comment 1: The authors propose a decentralized task delegation strategy called PATRA. They performed a comprehensive evaluation of the different characteristics present in the design.
Unfortunately, I am missing a comparison with other strategies to solve the same problems. Thus, the authors fail to answer whether PATRA makes sense as a strategy, especially for the proposed use cases. For example, what is the team size for PATRA to have a significant impact? A centralized coordinator/dispatcher might allocate resources/agents better in an extensive search and rescue operation, while PATRA might be better for small teams. In addition, when we think about military strategies, for example, there is the idea of a squad leader, a person in the unit that has the best overview of the capabilities of each team member and usually has excellent situational and strategical awareness. So how does PATRA stack against the team leader model? Even when considering hybrid teams, the team leader model unburdens each agent from being fully aware of the other's capabilities. Furthermore, how is PATRA different from on the field problem-solving?
Comment 2: Regarding the choice of use cases, I'm not sure the component of explaining your capabilities in a critical situation is an adequate strategy. I am not sure there is time to update capability models in the field. You either can handle the situation or report it and move on. Therefore, I would encourage the authors to find use cases where PATRA makes sense.
Response: We agree that in some occasions the performance of the team is prioritized to team learning, especially in critical situations. We do however believe that the selected use cases contain relevant team-dynamic properties that are relevant for TDP PATRA. Furthermore, also critical situations are regularly exercised to become more efficient as a team. During exercises there is an explicit focus on team learning, which is enhanced by providing explanations of agents. In order to make this clearer in the manuscript, a new section has been added (4.2), which describes the use case characteristics and the applicability (and when it is no applicable) of PATRA in detail.
Comment 3: Furthermore, with the proposed use cases, the fact that the paper addresses hybrid teams is irrelevant. PATRA is a strategy for resource allocation in a problem-solving scenario. Whether the agent is a person or a robot becomes irrelevant.
Response: Thank you for this observation. We agree that PATRA is also applicable in human-human teams, or robot-robot teams. There is however a significant difference in hybrid teams.
Humans and agents have different information processing systems that underly their intelligence. Hence achieving common ground and aligning task objectives sets additional demands. We understand that this was not clear enough in the manuscript, and we have revised the introduction to bring this point across more clearly.
Comment 4: Finally, I would ask the authors to revise the paper to write more succinctly. Please compress the introduction and the discussion.
Response: Agree. We have revised the introduction and discussion to be more to the point.
Reviewer 3 Report
This paper introduce a Team Design Patterns (TDP) called “proactive task reassignment” (PATRA) to assign tasks more effectively among team members by utilizing the different strengths and weaknesses of humans and agents. Some detailed comments are as follow:
- This paper is not well organized. For example, there should be a section focusing on literature reviews of existing TDP methods.
- The contributions of this paper can be better organized, e.g. What are the existing ways to assign tasks among team members? What are the differences between the proposed approach and the existing ones? How does the proposed approach address the challenges of existing ones? It will be better if these discussions can be included in the revised manuscript.
- The paper writing could be improved: there are some grammar mistakes, and the sentences are not concise.
- Experimental validation can be improved: This paper only conducted the experiments for the proposed method without comparing with other existing TDP approaches.
Author Response
We appreciate the time and effort that you have dedicated to providing your valuable feedback on our manuscript. We are grateful for your insightful comments on our paper. We have been able to incorporate changes to reflect most of the suggestions. We have highlighted the changes within the manuscript.
Here is a point-by-point response to your comments and concerns.
Comment 1: This paper is not well organized. For example, there should be a section focusing on literature reviews of existing TDP methods.
Response: Agree. We have, accordingly, revised the introduction and created a literature section.
Comment 2: The contributions of this paper can be better organized, e.g. What are the existing ways to assign tasks among team members? What are the differences between the proposed approach and the existing ones? How does the proposed approach address the challenges of existing ones? It will be better if these discussions can be included in the revised manuscript.
Response: Thank you for pointing this out. The original manuscript may have suggested that our focus was on the design of PATRA, and that our aim was to test our design of PATRA. But that is not what we did. PATRA is just an example of how collaboration between humans and agents can be effectively organized. We selected PATRA as an example of a TDP and in a previous agent-agent study we demonstrated that PATRA can indeed improve collaboration and that it improves performance (somewhat). In order for a team to successfully apply PATRA, the team members must have a sufficiently elaborated mental model (for humans) or knowledge representations (for agents). Furthermore, we hypothesize that when explanations are provided in the execution of PATRA, this will enable team members to elaborate and refine their mental models (or representations) of the task, of the team members, and of themselves. We investigated this hypothesis, and focused this on one direction: we tested whether a human participating in a team in which team members provide explanations when applying PATRA, develops a better understanding (=mental model) than a human participating in a team in which team members just apply PATRA without explanations. Furthermore, it was tested whether adding explanations also affected over-all team performance. In order to make this clearer in the manuscript, the introduction and discussion was revised.
Comment 3: The paper writing could be improved: there are some grammar mistakes, and the sentences are not concise.
Response: We have proof-read the entire article and improved several language issues. We furthermore have revised the introduction and discussion to be more to the point.
Comment 4: Experimental validation can be improved: This paper only conducted the experiments for the proposed method without comparing with other existing TDP approaches.
Response: We agree that this would be a short-coming for an article with its focus on introducing a new team-design pattern. As mentioned in our response to your comment 2, our focus rather lies on determining the value of additional explanations of agents on the shaping of the human mental model of the human. We believe that the new version of the manuscript is clearer in that respect. Therefore we argue that the additional value of comparing PATRA to other TDPs is rather low. An interesting topic for future research in the line that you mention is what the effects of the TDP are on human-mental model shaping (with or without explanations). We have added it this as a potential future work.
Reviewer 4 Report
In this paper, the authors present PATRA, an approach to distribute the work associated with a task among team members. The presented work is interesting, easy to read, and sound. The approach and experimental setup have scientific merit and the results are well and clearly presented. Therefore, I will suggest accepting the paper after minor revision.
My main concern is the way the paper is structured. After the Introduction Section, I would suggest including a Related Work Section. This is already partially covered in the Introduction, however, it's probably good to keep this in different sections and include new works. Additionally, I would have expected to have the full definition of PATRA before the Use Cases Section, in order to better contextualize. Finally, I would split Discussion and Conclusion Section, or at least I would include subsections if authors decide not to split. If subsections are used, consider adding a subsection for Future Work too.
Please check [1, 2, 3] as they can be useful for related works or to improve the discussion and future work ideas.
[1] Dazeley, R., Vamplew, P., Foale, C., Young, C., Aryal, S., & Cruz, F. (2021). Levels of explainable artificial intelligence for human-aligned conversational explanations. Artificial Intelligence, 299, 103525.
[2] Bignold, A., Cruz, F., Taylor, M. E., Brys, T., Dazeley, R., Vamplew, P., & Foale, C. (2021). A conceptual framework for externally-influenced agents: An assisted reinforcement learning review. Journal of Ambient Intelligence and Humanized Computing, 1-24.
[3] Bignold, A., Cruz, F., Dazeley, R., Vamplew, P., & Foale, C. (2022). Human engagement providing evaluative and informative advice for interactive reinforcement learning. Neural Computing and Applications. https://doi.org/10.1007/s00521-021-06850-6
Author Response
We appreciate the time and effort that you have dedicated to providing your valuable feedback on our manuscript. We are grateful for your insightful comments on our paper. We have been able to incorporate changes to reflect most of the suggestions. We have highlighted the changes within the manuscript.
Here is a point-by-point response to your comments and concerns.
Comment 1: In this paper, the authors present PATRA, an approach to distribute the work associated with a task among team members. The presented work is interesting, easy to read, and sound. The approach and experimental setup have scientific merit and the results are well and clearly presented. Therefore, I will suggest accepting the paper after minor revision.
My main concern is the way the paper is structured. After the Introduction Section, I would suggest including a Related Work Section. This is already partially covered in the Introduction, however, it's probably good to keep this in different sections and include new works.
Response: Agree. We have, accordingly, revised the introduction and created a literature section.
Comment 2: Additionally, I would have expected to have the full definition of PATRA before the Use Cases Section, in order to better contextualize.
Response: Thank you for this suggestion. We have considered switching the sections and introduce the use cases after the PATRA, and it makes a lot of sense to do so. We do however have an issue with Section 4.3 (in the numbering of the new version of the manuscript). This section shows that the TDP can be valuable in the Blanket Search use case. The value of the TDP is however not the focus of the presented research, as the value of explanations are. So the sections cannot be simply switched as knowledge of the Blanket Search is required to understand the value of the TDP (4.3). The alternative would be to first introduce the TDP, then the use cases, then discuss the value of the TDP for the blanket search, and then continue with the main focus of the article. This structure however seems more confusing to the reader than the current setup. We therefore chose to maintain the current structure, but include a new section 4.2 that specifically discussed the value of PATRA in the earlier use cases to help the reader to contextualize the pattern.
Comment 3: Finally, I would split Discussion and Conclusion Section, or at least I would include subsections if authors decide not to split. If subsections are used, consider adding a subsection for Future Work too.
Response: We agree that the discussions and conclusions were somewhat intertwined. In order to make the distinction clearer, we have restructured the conclusions sections and applied subsections, including for future research.
Comment 4: Please check [1, 2, 3] as they can be useful for related works or to improve the discussion and future work ideas.
Response: Thank you for these references. We have added them and related works from them in the related research section of the manuscript.
Round 2
Reviewer 2 Report
I thank the authors for their effort. But I feel they have not sufficiently modified the paper to answer the requested changes.
I think the spirit of the previous revision still holds: the authors must compare their work against state of the art. Currently, references are adequate for a book-style publication but not for a scientific paper. We are missing the research space definition. Where do you improve with respect to the state of the art?
The authors might benefit by creating a list of takeaway messages and showing the results that support that message. The takeaway messages should answer to research questions stated in the introduction, which are currently not clearly defined. The paper's lack of focus is evidenced in section 4, where there are already conclusions based on results. Still, section 5 is the evaluation section, presenting more results. If the paper's contributions PATRA, I expect graphs, tables, and figures to show how PATRA (as a whole) performs better than any other methodology. Currently, the paper's main results seem to confirm the state of the art and not the advantages of PATRA.
I wonder whether the HALL reference is inadvertently distracting from the paper's goal. The problem with HALL was that the agent's underlying objectives/intrinsic motivations no longer were aligned with the team's objectives. Therefore, HALL had no reason to explain his reasoning. As far as I can tell, PATRA does not prevent this situation from happening. Consequently, I suggest using HALL only once in the introduction to motivate the paper. Further inclusion only muddles the paper's focus as it forces you to explain that HALL was not interested in explaining anything he was doing.
Please use the discussion section to elaborate on methodology limitations and the possible effects of any assumptions. The explanations provided in section 6 should be part of the description of the results or skipped altogether.
Please be concise overall.
Author Response
We appreciate the time and effort that you have dedicated to providing detailed comments on the new version of our manuscript. We are grateful for your sharp eye for issues that indeed were not sufficiently discussed in the manuscript. We have made changes accordingly and have highlighted the changes within the manuscript.
Here is a point-by-point response to your comments and concerns.
Comment 1: I think the spirit of the previous revision still holds: the authors must compare their work against state of the art. Currently, references are adequate for a book-style publication but not for a scientific paper. We are missing the research space definition. Where do you improve with respect to the state of the art?
Response: We agree with the reviewer that embedding our work in the context of the literature is important, and that the relevance of the study to the field should be clear. Your comments have pointed out to us that we need to be more precise. We have modified the last two paragraphs of the introduction to better explain our objectives. In the text of our previous version of the paper we may have inadvertently suggested that our paper is about the design of a new TDP, and to investigate whether it yields better effects than other TDPs. However, this is not the case (!), and the comments of reviewer 2 made us aware of this misconception.
The objective of our study is to argue that human-agent teams need carefully designed team design patters to guide the team into effective operation. The field of human-agent teaming needs empirical evidence for the claimed beneficial effects of TDPs. PATRA is a representative example of such a team design pattern, and in the pilot study we investigated and demonstrated that this TDP indeed positively affects team functioning and team performance. In the subsequent main study we investigated whether explanations by agent team members, when executing PATRA, support the human team member to improve its understanding of the task and the team. The results of our study contribute to the field of human-agent teaming as it provides empirical evidence for TDPs in a simulated, yet representative setting. Furthermore, it provides clues as to what elements (i.e., explanations) are important to include in a TDP in order to achieve effective team performance. We hope to have formulated the objectives of our study, and how it contributes to the field more clearly in the text of the revised version of the paper.
Comment 2: The authors might benefit by creating a list of takeaway messages and showing the results that support that message. The takeaway messages should answer to research questions stated in the introduction, which are currently not clearly defined. The paper's lack of focus is evidenced in section 4, where there are already conclusions based on results. Still, section 5 is the evaluation section, presenting more results. If the paper's contributions PATRA, I expect graphs, tables, and figures to show how PATRA (as a whole) performs better than any other methodology. Currently, the paper's main results seem to confirm the state of the art and not the advantages of PATRA.
Response: Based on your comment, we affirm that section 4.3 was insufficiently introduced and placed in context with the remainder of the article, leading to that readers may experience a lack of focus of the article. We have added a new introduction to this section that makes it clear to the reader why this experiment is important for the article, and also why it is not grouped with the main experiment in Section 5.
The purpose of this experiment is to investigate whether the pattern improves team performance in the selected use case, when compared to teams that do not exchange tasks. Testing the effectiveness of the TDP is important, as in the main study (Section 5) we investigate whether adding the element of "providing explanations" to the TDP leads to a better understanding in the human agent, and whether it yields additional effects on the team’s functioning and performance.
Comment 3: I wonder whether the HALL reference is inadvertently distracting from the paper's goal. The problem with HALL was that the agent's underlying objectives/intrinsic motivations no longer were aligned with the team's objectives. Therefore, HALL had no reason to explain his reasoning. As far as I can tell, PATRA does not prevent this situation from happening. Consequently, I suggest using HALL only once in the introduction to motivate the paper. Further inclusion only muddles the paper's focus as it forces you to explain that HALL was not interested in explaining anything he was doing.
Response: After re-reading our manuscript we agree with the reviewer and have removed the second mentioning of HAL9000 and only mention the reference once in the introduction.
Comment 4: Please use the discussion section to elaborate on methodology limitations and the possible effects of any assumptions. The explanations provided in section 6 should be part of the description of the results or skipped altogether.
Response: We agree that we had insufficiently discussed the limitations of our chosen methodology and we have added the new subsection 6.2 (new numbering) that discusses these limitations. When considering to merge the (previous) content with Section 5, as you mention, we noticed that the discussion of results of TDP PATRA are indeed misplaced. This was a preliminary experiment for measuring the effects of the explanations which was the main study. Section 6 should only discuss the main study, and we therefore have removed Section 6.1 (old numbering). In this setup the focus of the entire section, and the overall article, becomes much clearer. The removal also makes the article more concise.
Reviewer 3 Report
Although some of the comments are not fully addressed, but I think the paper quality is greatly enhanced after revision. I have no more comments for this paper.
Author Response
Thank you for reviewing the new version of our manuscript and your kind words.